# Research on blockchain smart contract technology based on resistance to quantum computing attacks

**Xinhao Zheng** *

Beijing Police College, Beijing, China

* noctis428@outlook.com

## Abstract

In recent years, blockchain technology has developed rapidly and has been widely used in medical, financial, energy and other fields. However, in the process of practical application, each blockchain is a small independent ecosystem, with all transactions and operations limited to the chain, resulting in a large number of mutually heterogeneous to independent blockchains. It presents challenges for cross-chain interactions, cross-organization data sharing, and cross-blockchain expansion, and hinders the wider application of blockchain technology. In addition, the traditional digital signature method based on elliptic curve cipher faces the threat of being cracked by quantum computing attacks. To solve the aforementioned problems, this paper proposed a blockchain smart contract technique based on quantum computing attack resistance(BSCTQCAT). The technique first introduces the digital signature of the lattice cipher into the blockchain to resist the quantum search algorithm attack. Then, based on the smart contract authentication scheme, the nodes on multiple heterogeneous chains are organized into an identity agent layer P2P network, through which transactions on the chain will establish a credible identity management and message authentication mechanism between different chains, solving the current problem that each chain is difficult to communicate with each other. In this paper, the performance of the algorithm is evaluated by simulating the Bitcoin transaction scenario and analyzing the experimental data.

## 1 Introduction

Blockchain is a distributed ledger technology whose decentralized nature brings new opportunities and challenges to many fields [1]. However, since each blockchain platform has its own unique design and rules, interoperability between different platforms has become a problem that plagues the development of blockchain. Therefore, realizing cross-chain communication has become one of the essential topics in current blockchain research [2–4].

Quantum computing is a type of computation based on the principles of quantum mechanics, which has more powerful computational capabilities than traditional computing [5]. It utilizes the superposition and entanglement properties of quantum bits to accelerate the

**Data Availability Statement:** All relevant data are within the paper and its Supporting Information files.

**Funding:** The author(s) received no specific funding for this work.

**Competing interests:** The authors have declared that no competing interests exist.

computational process by processing large amounts of data simultaneously in some cases [6–9]. Smart contracts, which are programmatically defined contracts that automatically enforce the terms of the contract and ensure the reliability of the transaction, are a core component of blockchain technology [10]. Smart contracts are executed on the blockchain without the intervention of a third party, enabling automation and reducing the risk of potential fraud [11–13].

However, with the development of quantum computing technology, existing digital signature algorithms, such as elliptic curve cryptography, lack resistance to quantum computing attacks [14, 15]. Therefore, the research motivation of this paper is to address the cross-chain communication, data sharing, and security challenges faced by current blockchain technologies.

Therefore, this paper proposed a blockchain smart contract technique based on resistance to quantum computing attacks(BSCTQCAT). The technique employs a digital signature algorithm based on lattice cipher to enhance the resistance against quantum search attacks. By introducing the lattice cipher, more robust cryptographic protection is provided to ensure data security. Meanwhile, a smart contract authentication scheme is constructed to organize nodes on multiple heterogeneous chains by establishing a P2P network with an identity agent layer. According to the transactions on the chain, a trusted identity management and message authentication mechanism is established, enabling reliable communication between different chains. With this technique, the problem of compartmentalization between different chains in the blockchain ecosystem can be effectively solved and more secure and reliable blockchain smart contract services provided. The main contributions of this paper include:

1. In this paper, a digital signature algorithm based on lattice ciphers is used to improve the resilience against quantum search attacks. The algorithm utilizes the characteristics of lattice ciphers to provide more vital cryptographic protection to ensure the security of smart contracts and blockchain data, effectively countering the risk of quantum computing attacks on existing digital signature algorithms.

2. In this paper, a smart contract authentication scheme is constructed to ensure the reliability of authentication during the execution of smart contracts through a trusted identity management and message verification mechanism. This improves the execution credibility of smart contracts and reduces potential fraud risks.

3. In this paper, for the interoperability problem between blockchain platforms, a solution is proposed to realize the organization of nodes on multiple heterogeneous chains by constructing a P2P network with an identity agent layer, which effectively solves the problem of compartmentalization between different chains in the blockchain ecosystem.

4. In this paper, through the proposed technology based on anti-quantum computing attacks, this paper successfully solves the risk of quantum computing attacks faced by smart contracts and blockchain, providing a safer blockchain smart contract service. This is of great significance in promoting the development and application of blockchain technology.

The remainder of this paper will be organized as follows. Section II will review prior work related to blockchain smart contract technologies resistant to quantum computing attacks and assess the strengths and weaknesses of existing schemes. Section III will detail the related technology and research background. Section IV will focus on the design of quantum-resistant blockchain fast signature algorithms, including the principles of the core algorithms and specific implementations to ensure protection against possible future quantum computing attacks. Section V will discuss in detail the implementation scheme based on smart contracts, including the writing of smart contracts, and the implementation of the authentication

scheme. Section VI will conduct performance evaluation and comparison experiments to verify the feasibility and efficiency of the proposed technique. Section VII will discuss the application effects and advantages of the proposed techniques in real-world scenarios by analyzing real-world application scenario cases. Finally, the full paper will be concluded and the direction of future research will be envisioned.

## 2 Related work

Quantum computing, as a new information processing technology, is considered a candidate capable of realizing a paradigm shift. In their research, Alghadeer et al. [16] introduced a quantum computer simulator called Pistrum based on a generalized gate model implemented on classical hardware. The Pistrum simulator is designed with the goal of simulating and debugging a variety of applications through the use of quantum circuits, with the addition of quantum noise limiting the coherence of the quantum circuits as appropriate. Fiore et al. [17] designed a modular blockchain software architecture that has the ability to coexist with multiple cryptographic algorithms. The purpose of this design is to be able to mine and add new quantum-secure blocks while retaining verified, quantum-cracked blocks. Easttom et al. [18] discussed that blockchain technology is already widely used in society and has been adopted by many industries. However, with the advent of quantum computing, current blockchain implementations are under threat. This is because most widely used blockchain implementations rely on several cryptographic algorithms that are known to be vulnerable to quantum computing attacks. In addition, Holcomb et al. [19] redesigned Hyperledger Fabric's credential manager and related specifications to integrate the concept of hybrid digital signatures. This design uses a combination of a classical signature and a quantum-secure signature to prevent both classical and quantum attacks. On this basis, Nilesh et al. [20] designed a practically realizable full-quantum blockchain model based on a generalized Gram-Schmidt procedure utilizing dimensionality enhancement. This model stores transaction information in multi-quantum bit states and encodes using the generalized Gram-Schmidt procedure.

Blockchain smart contracts are very common today as many applications are developed based on this feature. Despite the importance and widespread use of smart contracts, they may contain some vulnerabilities. Pise et al. [21] proposed some security measures that should be taken when writing smart contract code to prevent potential attacks from occurring in blockchain-distributed applications. Wu et al. [22] proposed three smart contract techniques through which smart contracts are customized and developed for the electric power business. These techniques ensure the security compliance of the business and automate and intelligentize the power of the business process, thus improving its efficiency. Security vulnerabilities in smart contracts are indeed an issue that has attracted widespread attention in recent years, as these vulnerabilities can lead to significant economic losses and risks. Zhao et al. [23] proposed a method to detect vulnerabilities in smart contracts of open blockchain systems and studied the current mainstream smart contract vulnerability detection methods. In addition, Abuhashim et al. [24] investigated the indexing and querying of carpooling data based on smart contracts. They proposed two smart contract designs, Catalog and Sparse smart contracts, to support indexing and retrieving blockchain data. Building on this foundation, Ren et al. [25] proposed a framework called CLOAK for the development of confidential smart contracts. The critical feature of the CLOAK framework is that it allows developers to implement and deploy practical solutions for the multi-party transaction problem. The framework handles the secret inputs and states owned by different participants by simply specifying the way.

In the above research, existing digital signature algorithms, including elliptic curve encryption, face the risk of being cracked by quantum computing attacks. In the event that a quantum

computing attack is able to successfully crack these algorithms, traditional digital signature mechanisms will no longer be secure. This issue will also have an impact on smart contracts, which typically use digital signatures to verify and authorize transactions. If quantum computing attacks crack traditional digital signature algorithms, the security of smart contracts will be jeopardized. Attackers potentially may be able to forge digital signatures or tamper with the execution results of smart contracts. Therefore, this paper proposed a blockchain smart contract technology based on anti-quantum computing attacks, which can effectively deal with quantum computing attacks and protect the security of smart contracts by adopting quantum-secure encryption algorithms and digital signature algorithms. In this way, the risk of quantum computing attacks is reduced, the security of digital signatures is ensured, which is not easy to be cracked or forged by attackers.

## 3 Technology background

### 3.1 Verifiable delay function (VDF)

**3.1.1 VDF definition.**   A verifiable delay function is a function f: X → Y that requires a specified wall clock time to compute even on a parallel processor and outputs a unique result that is effectively verified [26]. Even when evaluated on a massively parallel processor, the function still has to be evaluated in a series of consecutive steps [27]. Most importantly, given input x and output y, anyone can quickly verify that the output result y = f(x) is correct, where f: = X→Y satisfies the following requirements:

$$x \rightarrow x^2 \rightarrow x^{2^2} \cdots x^{2^T} (\bmod N) = y \tag{1}$$

The design of verifiable delay functions takes into account two key elements: delay and verifiability [28]. Latency refers to the time required to complete the computation, while verifiability refers to the verifiability of the output result. By designing functions that satisfy these two elements, verifiable delay functions are used in distributed applications in scenarios where time delays need to be added [29].

A verifiable delay function is usually defined as a tuple $(N, x, T)$, where N is the input domain of the function, $x$ is the input value, and $T$ is the number of iteration steps. For each input tuple, the function outputs a unique result $y$, which can be verified as the correct output of the function $f(x)$ under publicly verifiable conditions, as shown in Eq (2):

$$\mu := 2^T (\bmod \varphi(N)), y := x^\mu (\bmod N) \tag{2}$$

Where $N$ is the RSA modulus, which consists of the product of two large prime numbers $p$ and $q$, i.e., $N = p * q$, $x$ is a random seed belonging to $Z_N^*$, denoting a random value located in the range of modulus $N$. $T$ is the time delay parameter, which is a non-negative integer. In addition, $\varphi(N) = (p\text{-}1)(q\text{-}1)$ is an Eulerian function to denote the order of modulus $N$.

In addition, the triple algorithm for the verifiable delay function consists of three steps of Setup, Eval and Verify as shown in Fig 1 Algorithmic flow of the verifiable delay function.

Setup(λ, T) → pp: = (ek, vk) is a stochastic algorithm with inputs the delay parameter T and the security parameter λ, outputs the public parameter pp consisting of the evaluation key parameter *ek* and the authentication key parameter *vk*. The algorithm is under default assumptions required to guarantee a certain security strength.

For avoiding the impact on security, the running time of the algorithm should be limited to an acceptable range. Thus, a suitable running time based on the security parameter λ has to be chosen [30]. Also, choosing random numbers is an important issue, which should be secret and sufficiently random to ensure security.

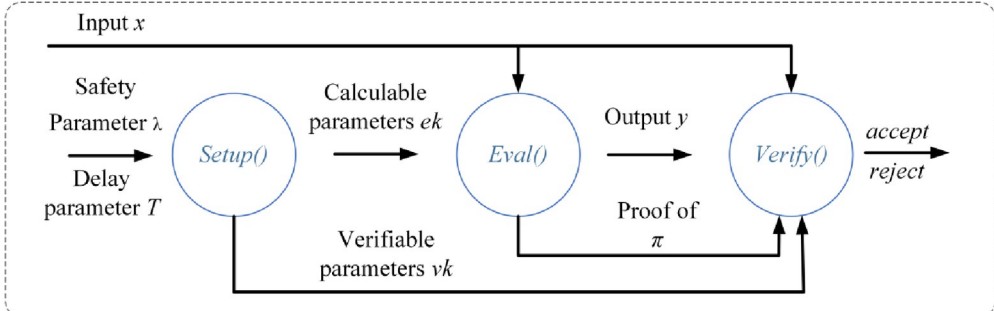

**Fig 1. Algorithmic flow of the verifiable delay function.**

Eval($ek$, $x$) → ($y$, $\pi$) is a slow cryptographic algorithm that accepts as input the evaluation key parameter ek and a random seed $x$ and outputs the result y along with a possibly null proof $\pi$. In order to ensure sequentiality, the *Eval* algorithm has to complete in a specified amount of time and needs to satisfy the requirements of a polynomial logarithmic parallel processor.

Verify($vk$, $x$, $y$, $\pi$) → {*accept*, *reject*} is a deterministic cryptographic algorithm that accepts as input a verification key parameter $vk$, a random seed $x$, an output value y, with a possibly null proof $\pi$. The Verify algorithm is a simple, simple, and easy to use algorithm. It outputs *accept* if $y$ is equal to $f(x)$ and *reject* otherwise. The *Verify* algorithm runs much faster than the *Eval* algorithm, which takes a total time polynomial in $\log(T)$ and $\lambda$ to complete its execution.

**3.1.2 VDF in blockchain.**  Existing centralized timestamping service methods suffer from a single point of failure in the blockchain structure, where users need to assume that the time-stamping server is secure and trustworthy [31]. In contrast, timestamping data on the chain can be overly burdensome for users. In addition, the existing distributed methods must provide timestamps through a set of voluntary issuers. Users require real-time interactions with all issuers to guarantee the security of outsourced data. This not only increases the communication burden of users but also increases the cost overhead of all issuers [32].

To solve the above problems, a blockchain-distributed timestamping model architecture with a continuously verifiable delay function can be used, as shown in Fig 2 Distributed time-stamping model for blockchain based on continuously verifiable delay function, which does not rely on any third-party trusted entity and uses the blockchain-distributed network for storage.

The model is divided into three main layers: hash calculation service layer, distributed time-stamp service layer, and blockchain layer.

Hash calculation service layer: The main function of this layer is to provide hash calculation for the input data and return its hash value. In this layer, existing hash functions or hash algorithms can be used to complete the calculation without writing their hash functions or implementing hash algorithms [33]. Hash computing services have a very important role in data security, encryption, and integrity verification, which helps users achieve data security and integrity assurance [34].

Distributed Timestamp Service Layer: This layer adopts a precise time source and a high-intensity and high-standard security mechanism, which is used to confirm the existence of the data processed by the system and the relative time sequence of the relevant operations, providing a basic service for the prevention of time repudiation and tampering in the information system [35]. The timestamp server is a timestamp authority system based on public key infrastructure technology, providing accurate and trustworthy timestamp service for all users. In the past, the scheme of providing timestamp service to data through a single timestamp server

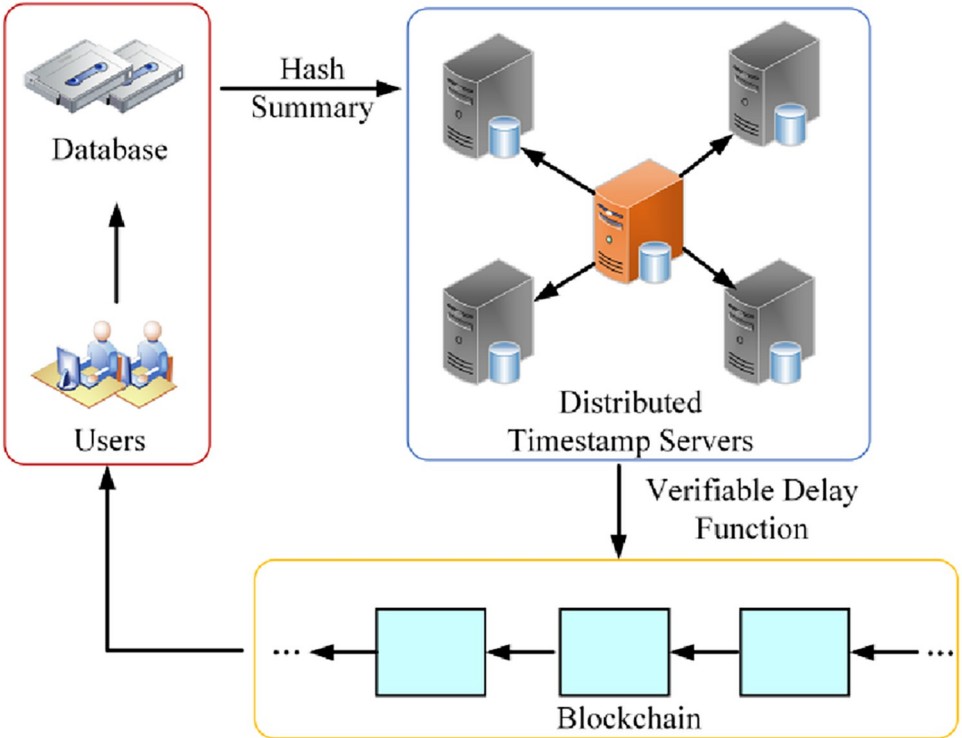

**Fig 2. Distributed timestamping model for blockchain based on continuously verifiable delay function.**

easily led to collusion between users and timestamp service providers, failing to ensure the security of data adequately [36]. Therefore, the distributed timestamping service layer adopts the blockchain distributed timestamping scheme based on a continuous verifiable delay function to provide timestamping service, and maintains the timestamps of the data through multiple timestamping servers together. In this process, the main timestamp server is responsible for providing a time synchronization service for the whole system.

Blockchain Layer: This layer is the core of the whole model, which utilizes the characteristics of blockchain to store and protect timestamp information. The decentralized feature of blockchain can prevent a single point of failure and ensure the availability and stability of timestamp service [37]. In addition, the transparency and non-comparability of the blockchain can also ensure the trustworthiness and security of the timestamp [38].

## 3.2 Smart contract

**3.2.1 Definitions and functions.** A smart contract is a protocol that digitizes and computerizes contractual information, which is disseminated and verified with the operating mechanism of a computer [39]. The essence of a smart contract is a protocol executed on a computer that has the ability to program logic and to logicize physical protocols. Smart contracts translate the conditions, rules and constraints of a contract into executable code through a programming language and a computer execution environment [40]. This code is deployed on a distributed system, such as a blockchain, where the content of the contract is verified and enforced through the consensus of the participants.

Smart contracts do have some decentralized and tamper-proof features compared to traditional contracts. First, smart contracts incorporate blockchain technology and are no longer

subject to the will of the parties involved [41]. Trigger conditions are set through programming code, and once the conditions are met, the contract takes effect until the event is finished. The immutability of the blockchain ensures that once a smart contract is predefined, no changes are allowed. The smart contract is verified by miners and packed into the block, waiting to be triggered. Then, smart contracts are decentralized [42]. Participants meet the rules in the smart contract and trigger the program in the smart contract to run without the involvement of a third-party trust organization. Smart contracts are recognized and verified through consensus protocols in the blockchain, will not be tampered with once confirmed, and are trusted by nodes in the entire distributed network [43]. The de-will feature of smart contracts makes contract execution more reliable and trustworthy, reducing the intervention of intermediaries and trust costs, which can be applied in various scenarios, such as financial transactions, supply chain management, digital asset transactions, etc., providing a more efficient, reliable and secure contract execution environment through the characteristics of blockchain [44].

**3.2.2 Security considerations for smart contracts.** The security of smart contracts is a crucial aspect of a smart contract system that ensures reliability, confidentiality, and tamper-proofing during the execution and use of the contract [45]. The following are some of the important considerations and measures for the security aspect of smart contracts.

Programming security: Smart contracts should be written following secure coding practices to prevent potential vulnerabilities and attacks. This includes avoiding obsolete or risky functions, using correct data types and algorithms, performing input legitimacy checks, etc [46]. The code of a smart contract should be carefully reviewed and tested to ensure no logical errors or security vulnerabilities.

Access Control and Privileges: Smart contracts should limit the access and operational privileges of the contract based on specific business requirements and access control policies [47]. For contracts involving sensitive data or critical operations, there should be a clear permission management mechanism so only authorized users can perform the corresponding operations [48]. Key functions and data in the contract should be appropriately access-controlled to ensure that only legitimate users can invoke and modify them.

Input validation and exception handling: Smart contracts should validate and handle various inputs during execution to prevent security problems caused by abnormal or malicious inputs of input data [49]. This can include verifying the integrity, legitimacy and scope of inputs, as well as handling potential anomalies and errors. Contracts should have good error-handling mechanisms to catch and handle anomalies to prevent the contract from being abused or attacked [50].

Preventing Reentry Attacks: Reentry attacks are malicious contracts that exploit the vulnerability of contracts calling each other. To prevent reentry attacks, smart contracts should follow secure calling patterns and use appropriate locking and state management mechanisms to avoid reentry situations [51].

Secure Auditing and Public Verifiability: Smart contracts should be auditable and verifiable so that third-party independent organizations or users can audit and verify the behavior and security of the contract [52]. This can be achieved through the transparency of the contract code, the traceability of the contract history, and the consensus of the nodes on the blockchain. Security auditing and access review of smart contracts are important parts of ensuring the security of the contract and can be implemented through security auditing tools and third-party audits.

Updating and upgrading smart contracts: Once a smart contract is deployed and verified, its code and logic are locked on the blockchain and cannot be changed [53]. However, it is a challenge to update and upgrade contracts that have vulnerabilities or need improvement. A

common practice is to manage contract updates by setting up upgrade contracts or using proxy contracts while ensuring security and legality.

Therefore, the security of smart contracts needs to consider a combination of programming security, access control, input validation, exception handling, anti-re-entry attacks, auditability, and contract upgrades [54]. A secure smart contract needs to be thoroughly designed, tested and audited, combined with reasonable technical and management measures to ensure its security in practical applications.

### 3.3 Quantum computers and anti-quantum cryptography

**3.3.1 Working principles of quantum computers.** A quantum computer is a computer that uses a quantum bit (qubit) as the unit of computation. Unlike the classical bits (bits) used in conventional computers, quantum bits are allowed to be in more than one state at a time, thus providing greater computational power and the potential to break codes [55].

The best feature of a quantum bit is its quantum superposition state. While a bit in a conventional computer can only be in two forms, 0 or 1, a quantum bit can be in a superposition of 0 and 1 at the same time. This is due to the fact that quantum bits utilize the principle of superposition in quantum mechanics, i.e., a quantum system can be in multiple states at the same time, and these states can interfere and superpose each other [56]. The representation of a superposition of states is usually described mathematically by a wave function, where the magnitude indicates the weights of the different forms.

Another important property of quantum bits is quantum entanglement. Quantum entanglement is the existence of a special correlation between two or more quantum bits such that changing the state of one quantum bit immediately affects the states of the other entangled quantum bits. This correlation is non-classical and is not limited by spatial distance; even if two quantum bits are far apart, the entanglement between them still exists [57]. Quantum entanglement is an important resource in quantum computing that can be used to realize quantum communication and quantum information processing.

In quantum computing, the processing of information is realized through quantum gates. A quantum gate is a reversible operation acting on one or more quantum bits that enables the transformation of quantum states. The most common quantum gates include Hadamard gates, Pauli gates and CNOT gates. By appropriately combining and controlling the operation of quantum gates, quantum computers can realize parallel computation and quantum parallel algorithms, thus dramatically improving computational efficiency.

**3.3.2 Key points of anti-quantum algorithms.** Anti-quantum algorithms are a class of algorithms designed to cope with the emergence of large-scale quantum computers in the future. In traditional computers, public key cryptographic algorithms, hash functions and symmetric key algorithms are used to secure data [58]. However, with the development of quantum computers, these traditional cryptographic algorithms will soon lose their security.

The anti-quantum algorithm is mainly a new type of encryption algorithm designed to cope with the cracking attack of quantum computers [59]. Its basic principle is to utilize the shortcomings of quantum computers to design an encryption algorithm that is difficult to crack [60]. Here are the key points of the anti-quantum algorithm:

Quantum Random Number Generation: A quantum random number generator is a device that generates true random numbers based on quantum physical phenomena. These true random numbers are generated with a high degree of randomness and are not affected by quantum computer attacks. Quantum key-resistant encryption and digital signature algorithms can utilize quantum random number generators to generate random numbers for encryption and signatures [61].

Optical encryption: Optical encryption utilizes the physical properties of optics to convert signals into light signals for transmission, thus avoiding the weakness of electronic signals that quantum computers can crack [62]. Optical encryption algorithms can rely on phenomena such as quantum entanglement and superposition states to achieve encryption.

Lattice cipher-based encryption: A lattice cipher is a mathematically based encryption algorithm whose security relies on discrete mathematical problems on a multi-dimensional lattice [63]. Lattice ciphers are resistant to Shor's algorithm attacks on quantum computers because Shor's algorithm cannot efficiently deal with high-dimensional lattice problems.

Reinforced Hash Algorithm: A hash function is an algorithm that maps arbitrary length data to a fixed length output [64]. The reinforced hash algorithm uses a new hash function that does not rely on the mathematical principles of traditional hash functions for its security but is based on physical principles to achieve secrecy.

## 4 Design of fast signature algorithm for quantum-resistant blockchain

### 4.1 Design objectives and requirements

Traditional public key cryptosystems have taken a huge hit with the rapid development of quantum computers, as large integer factorization and discrete logarithm problems become solvable on quantum computers [65]. This also means that the elliptic curve-based digital signature regimes used in blockchain are no longer secure against quantum computing attacks [66]. In addition, researchers have pointed out that the user's identity information in blockchain transactions is "pseudo-anonymity", which is a problem that plagues the development of blockchain technology. Therefore, this paper proposes a fast certificate-less signature algorithm based on lattice to solve the problem of user identity privacy leakage in blockchain transactions [67].

### 4.2 Infrastructure framework

The overall construction of user identity privacy and security against quantum computing attacks security question in the post-quantum blockchain anonymous transaction scheme constructed using the The Lattice Cryptography-based Certificate-Less Rapid Signature (LWE-CLRS) algorithm is shown in Fig 3. Architecture of fast signature algorithms for quantum-resistant blockchains.

As shown in the figure, the user needs to send a transaction request to the supplier who provides the resources for this transaction to make a transaction with him. To ensure the security and privacy of the transaction, the LC-LRS algorithm is utilized for ring signing. First, the user requests the public parameters and a portion of his/her public-private key pair from the key generation centre (KG). The user has to determine the appropriate number of ring membership, the more number, the higher the privacy of the transaction, but the computational and verification complexity of the transaction will also increase. The user utilizes the LC-LRS algorithm to sign the transaction quickly and ensure the transaction's validity. Therefore, the user sends this transaction request to the supplier. Eventually, the supplier receives the transaction request, performs confirmation to verify whether the transaction is legitimate and writes the transaction record in the blockchain, thus completing the entire transaction process.

### 4.3 Detailed steps of the algorithm

This subsection details a fast certificate-less signature algorithm based on lattice cryptography (LWE-CLRS).

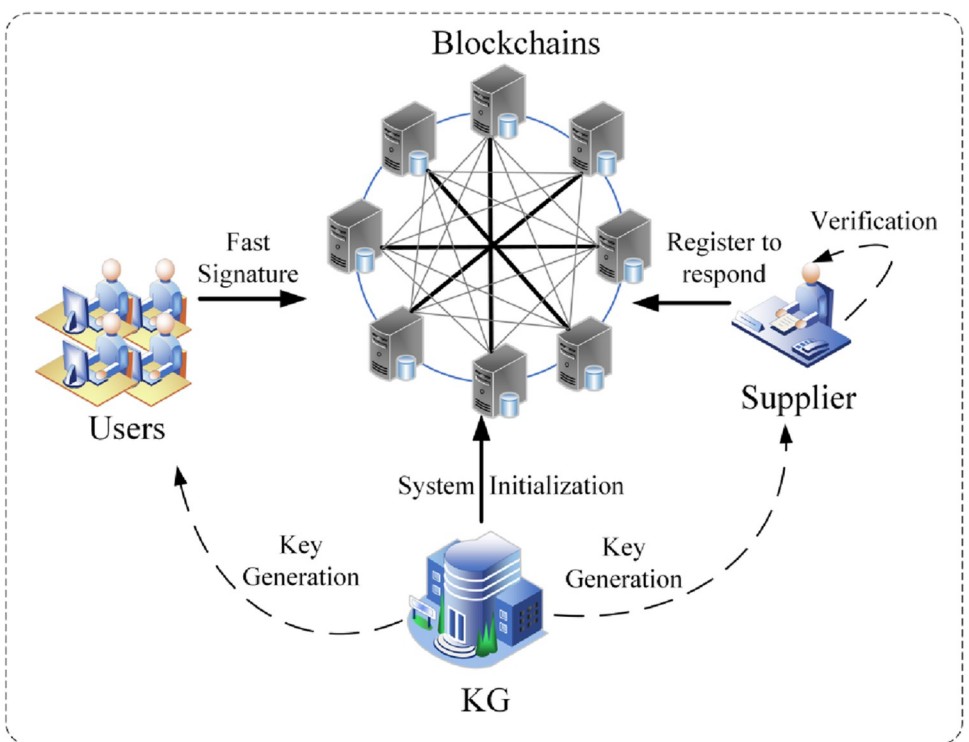

**Fig 3. Architecture of fast signature algorithms for quantum-resistant blockchains.**

System Initialization: The input parameters are $n$, a prime number $q$ greater than 3, and $m = 0\ (\sqrt{n\log q})$, $L > 0\ (\sqrt{n\log q})$, $0 \leq \sigma < \sqrt{n\log m}$. KG (Key Generation Center) generates the system parameters $P$ and the master key $MSK$ in the manner described in the System Initialization section. The system parameters $P$ consist of the public and key parameters, and the master key $MSK$ is used to generate the user's private key. The master key $MSK$ is used to generate the user's private key. Execute the trapdoor generation algorithm so as to output the matrix $A$ and lattice cipher $T_A$ as shown in Eq 3.

$$TrapGen(1^n) \rightarrow (A, T_A) \tag{3}$$

Next, three secure hash functions are selected, as shown in Eqs (4)–(6), and the final output is the public parameter $P$ and the system master key $MSK = T_A$.

$$H_1 : \{0, 1\}^* \rightarrow Z_q^{n \cdot k} \tag{4}$$

$$H_2 : \{0, 1\}^* \rightarrow \{v : v \in \{-1, 0, 1\}^n, \|v\| \leq t, t \in R\} \tag{5}$$

$$H_3 : \{0, 1\}^* \rightarrow Z_q^{m \cdot n} \tag{6}$$

Key Generation: Key generation needs to be performed in the $KG$, which generates the private key $sk$ and public key $pk$ for the user based on the public parameter $P$ and the master key $MSK$.

KG uses the matrix original image sampling algorithm when generating partial private keys for users with identity IDs. This algorithm is used to generate matrices that satisfy specific constraints to ensure that the generated private key meets the requirements of security and

certificate-free linkability, as shown in Eq (7).

$$sampleMat(A, T_A, \sigma, R) \rightarrow S'_{ID} \tag{7}$$

where $R \in Z_q^{n \cdot k}$, $S'_{ID} \in Z_q^{m \cdot k}$, and satisfies $AS'_{ID} = R$. then KG generates the user partial private key $S'_{ID}$, thus KG along with the selection matrix $A' \in Z_q^{n \cdot k}$, and sends the $S'_{ID}$ and $A'$ to the user.

The secret matrix $p \in Z_q^{n \cdot k}$ is randomly selected and saved by the user himself or generated and sent to the user by the KG.

Since $m > n$, if the user wants to generate a public-private key pair, randomly select the $n$ rows of the matrix $S'_{ID}$ and transpose to get the matrix, thus calculating $A''$, $A_{ID}$ and $S_{ID}$, as shown in Eqs (8)–(10).

$$A'' = A'S' + A'P(\bmod q) \tag{8}$$

$$A_{ID} = [2A'|2A'' + qI] \tag{9}$$

$$S_{ID} = [S' + PI| - I]^T \tag{10}$$

Where $A_{ID} \in Z_q^{n \cdot m}$, $S_{ID} \in Z_q^{m \cdot n}$, $A'' \in Z_q^{n \cdot n}$, $I$ is the unit matrix. The final computation yields $Q = A_{ID}S_{ID}(\bmod 2q) = qI$, which outputs the user's public key $A_{ID}$, and private key $S_{ID}$.

Fast Signature: Set the user of fast signature as $U = \{ID1, ID2, ID3,\ldots, IDl\}$, public key as $pk = \{A_{ID1}, A_{ID2}, A_{ID3},\ldots, A_{IDl}\}$, $l$ as the maximum number of members. $\mu$ is the surrogate signature message and $IDi$ expresses the signer identification already, therefore the formula for fast signature is shown in Eq (11).

$$Zi = 1 \Big/ \left(M\exp\left(-\frac{\|S_{IDi}c\|^2}{2\mu^2}\right)\cosh\left(\frac{<y_i, S_{ID}, c>}{\mu^2}\right)\right) \tag{11}$$

Verification: Input membership information $U$ and $pk$ are verified, if all $i \in [l]$, have $\|zi\| \leq 2\sigma\sqrt{m}$ and $c = H_2\left(\sum_{i=1}^{l} A_{IDi}Zi + qc(\bmod 2q), pk, \mu\right)$ then the condition holds to pass the verification, otherwise the fast signature is invalid.

## 4.4 Security analysis of algorithms

The LWE-CLRS algorithm is a state-of-the-art cryptographic signature algorithm whose security is based on the LWE difficulty problem, which is the difficulty of approximating the solution of a linear problem in a system of polynomial equations. The difficulty of the LWE problem has been studied and verified many times and is considered relatively secure.

The LWE-CLRS algorithm provides aggregation security and chaining security whereby the signatures of multiple signers are aggregated and guaranteed to be equivalently secure. This aggregation security is achieved by a rational scheme design and is related to the chain length of the signatures. Various secure aggregation schemes have been shown to be feasible, allowing the algorithm to provide high-strength security in practical applications.

The chain structure is one of the core features of the LWE-CLRS algorithm, which allows chain linking of ring signatures, thus enabling fine-grained authorization management and security guarantees. The security of the chain structure requires maintaining the relevance of the signatures and needs to have the aggregation security of the chain structure. Such a chain structure can provide more flexible signing and verification methods.

In addition, the LWE-CLRS algorithm supports revocability, which allows the signer to revoke a previously published signature without revealing the private key. This feature is

realized through the support of a revocation key, which needs to be kept private and unbreakable. Such a mechanism allows the algorithm to provide a secure solution in cases where a specific signature needs to be revoked.

## 5 Smart contract based implementation

The techniques in resisting quantum computing attacks can be utilized in the identity management of smart contracts to ensure the privacy protection of the identity.

### 5.1 Smart contract platform selection

Ether is a great platform to consider when choosing a smart contract platform. Ethernet is one of the most popular and widely used smart contract platforms, with an extensive developer community and a mature ecosystem of tools. Writing smart contracts using the Solidity programming language leverages Ethernet's vast ecosystem for complex distributed applications.

Second, Ethernet is one of the leading scalable smart contract platforms, offering multiple solutions to address network scaling. Layer-2 solutions, such as Rollups and Sidechains, can significantly improve network throughput and performance. In addition, EtherNet is also working on Ethereum 2.0, a new, more scalable blockchain platform that will dramatically increase EtherNet's performance and throughput.

Beyond that, Ether has many advantages. For example, it provides a wide range of development tools, debuggers and libraries, which can significantly improve the efficiency and quality of smart contract development. Ethernet also supports token issuance and management based on standards such as ERC-20, ERC-721, and ERC-1155, which means that assets can be transferred and exchanged more easily.

Additionally, Ether is committed to developing a community ecosystem that enables Ether smart contracts to interact with other Ether-compatible blockchains through EVM (Ether Virtual Machine) compatibility. This interoperability enables more application scenarios, such as leveraging the DeFi protocol, non-homogenized token (NFT) markets, etc.

### 5.2 Smart contract deployment

The identity management aspect can be done by the smart contract through the agent layer. When an identity (an identity on the chain or a user's identity) is registered, the smart contract can index it and perform identity logout or update operations. However, as the number of identities increases, the identity information table maintained by the agent layer becomes enormous, which makes the management of identity lookup more and more inconvenient.

For improving the efficiency of identity lookup, the Bloom filter algorithm can be considered. Bloom filter is a fast and efficient data structure for checking whether an element exists in a set. Using an array of bits along with several hash functions, it determines whether an identity exists or not by hashing the identity to several locations in the bit array and checking whether these locations have been set, as shown in Algorithm 1.

```
Algorithm 1: Trusted Identity Search
Input: BF = {0}ⁿ, identifier[m]→ Array of query identity collections.
id → Identity to be queried
  i = 0
  // Loop over the m identity sets identifier to be queried
  While(i<m){
    j = 0;
    // For each identity identifier[i], compute the hash value by hⱼ(i-
dentifier[i]) and set the corresponding BF array element to 1 to indi-
cate that the identity may exist.
```

```
 While(i< = k){
 BF[□_j(identifier[i])] = 1;
 j = j+1;}
 i = i+1;}
//Find id
For i in range(0, k+1):
   // Iterate through the BF array, if there exists any BF array ele-
ment of 0, it means the corresponding identity does not exist, return
False
   IF (h_i(identifier))! = 1:
     Return False
   //If all the BF array elements are 1, it means all the identities
to be queried exist, return True.
   Return True
```

## 5.3 Integration of smart contracts with signature algorithms

In the case of organizing nodes on multiple heterogeneous chains into an identity agent layer P2P network, a trusted identity management and message authentication mechanism is established between different chains using on-chain transactions. This is achieved through smart contracts, which are used to manage the identity certificates of the nodes on the identity agent layer as well as the blockchains they belong to, as show in Fig 4 Architecture diagram for the integration of smart contracts and signature algorithms.

First, a smart contract is created on each chain for identity management. This smart contract maintains the identity information of each node, including the identity book and the corresponding public key. When a node needs to join the identity agent layer network, it registers its identity through on-chain transactions. It stores its identity book and public key in the corresponding smart contract.

When inter-chain communication is performed, the agent layer nodes of the participating nodes will first perform inter-chain identity authentication. The authentication is performed through the identity information in the smart contract. For example, when user A needs to communicate with user B, the agent layer nodes of their respective chains query each other's identity information, including identity certificates and public keys, through the smart contract. The node then signs the message using the appropriate signature algorithm and performs message verification to ensure that the other user is from a legitimate blockchain network.

To enhance security, smart contracts are integrated with signature algorithms. In the smart contract, the relevant verification logic and signature algorithm are defined to ensure the authenticity and integrity of the message. In this way, when a node makes a transaction on the chain, the smart contract verifies and determines whether the transaction is legitimate or not. Meanwhile, the signature algorithm is used to sign the transaction and the message to ensure the identity of the sender of the message and the integrity of the message.

## 6 Performance evaluation and comparison

### 6.1 Experimental settings

The effectiveness of the scheme proposed in this paper is tested through simulation experiments to verify whether the scheme proposed in this paper is capable of meeting the resistance to quantum computation in cross-chain communication scenarios. The experiments in this paper are run on Ubuntu 21.10 operating system. During the experiments, the system is utilized to achieve the fusion of blockchain smart contracts that are resistant to quantum computing attacks, with the application configuration shown in Table 1.

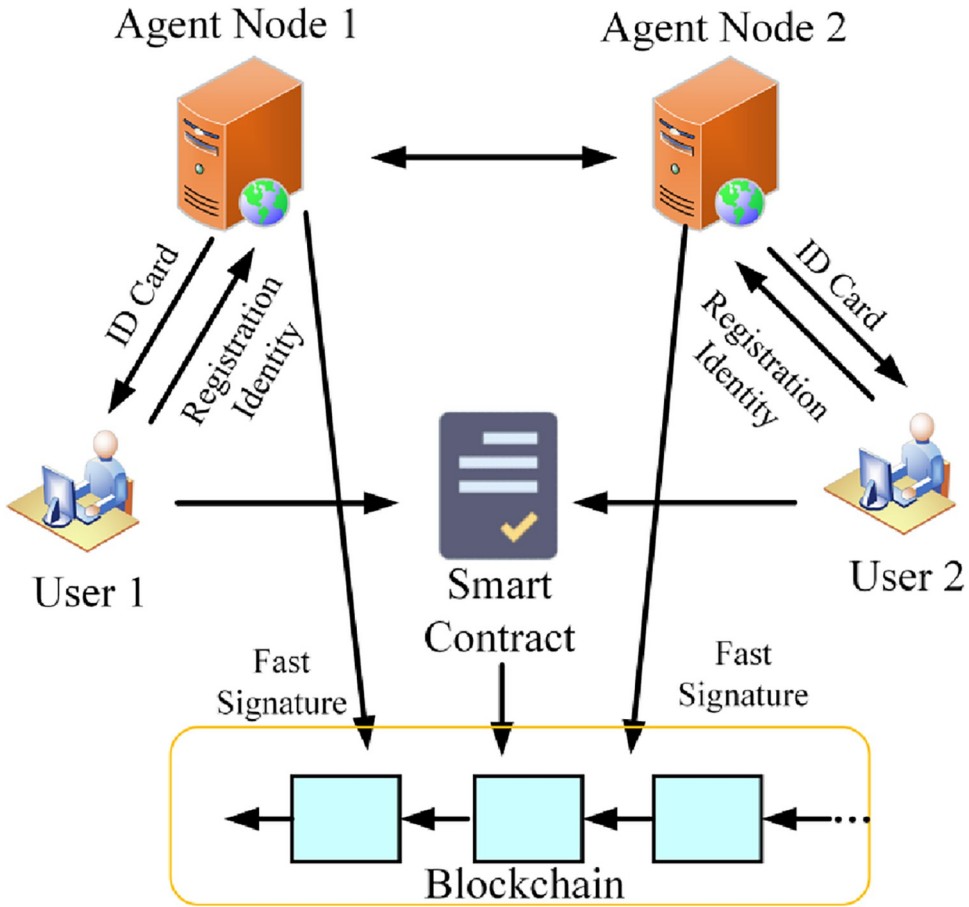

**Fig 4. Architecture diagram for the integration of smart contracts and signature algorithms.**

In this experiment, the following configurations and tools were used: an AMD Ryzen 9 5900HX processor was used for the CPU, and the RAM was 32 GB. For the operating system, Ubuntu 21.10 was chosen as the base platform. For the programming of smart contracts, the Solidity platform was used, while the writing of consensus algorithms was done on the Python platform. As the blockchain platform, HyperLedger Fabric was chosen. The above combination of configurations and tools will provide a stable and efficient operating environment for the experiments in order to facilitate the related simulation work.

## 6.2 Speed of each stage

In the simulation system in this paper, transactions are performed in a "serial" fashion. Whenever a transaction is created, a signature and a verification result are required. For each simulation, 1000 different experimental tests were conducted to obtain the durations of signatures,

**Table 1. Experimental environments.**

| Software Configuration | Parameters | Hardware Configuration | Parameters |
|---|---|---|---|
| CPU | AMD Ryzen 9 5900HX | Operating Systems | Ubuntu 21.10 |
| Hard Disk | 2TB | Development Languages | Python, Solidity |
| RAM | 32GB | Blockchain Platform | HyperLedger fabric |

**Table 2. Time for each phase of a digitally signed blockchain transaction.**

| Stage | Average time | Standard deviation of time |
|---|---|---|
| Signature | 0.035 | 0.003 |
| Verification | 0.184 | 0.004 |
| Transaction | 0.235 | 0.006 |

verifications, and transactions. This experimental design provides a more comprehensive understanding of the performance of the system for evaluation, as shown in Table 2.

According to the results in Table 2, the durations of each phase are all at the microsecond level, which means that the performance of the whole blockchain system system is less affected by the fact that the transactions are done in a "serial" way. However, the serial approach clearly shows the specific flow of each transaction, including the signature and verification process, which is easy to understand and debug. Each transaction is executed in a precise sequence without the complexity and competitive conditions associated with concurrent operations.

The result indicates that the algorithm is extremely efficient in processing transactions with fast response capability. Since the duration of each stage is very short, the entire transaction process can be completed quickly, thereby not creating a significant delay to the processing speed of other transactions. Thus, the current blockchain system has excellent efficiency in processing transactions.

### 6.3 Security assessment

Simultaneously, in this paper, malicious nodes that become KG nodes are tested. These KG nodes have a reputation value of 50 in the initial stage, which increases significantly as they continue to work honestly. However, if the KG nodes start to perform malicious behaviors, Fig 5 Results of the 50 rounds of voting on the evil KG node shows that their reputation value will drop dramatically. When the reputation value falls below a predetermined threshold, the KG node will be stripped of its identity and the system will initiate the process of re-voting for election.

The mechanism imposes severe penalties on nodes with high scores but evil behaviors, while rewarding nodes with well-intentioned behaviors handsomely. The entire blockchain encourages nodes to cooperate with each other for overall stable operation, while taking a firm stance against malicious nodes to create a safe operating environment.

### 6.4 Experimental comparison

In this section, the efficiency of the cross-chain communication system is tested in simulation comparison. In cross-chain communication, various entities interact with information, which mainly involves smart contracts and identity fast digital signatures on the blockchain. Owing to the redundancy of the blockchain system, each agent layer node backs up the identity data. However, an increase in the number of user node authentication and verification requests, or an increase in the number of blockchain networks, leads to more hashing operations by the agent layer nodes.

As shown in the Fig 6 Number of cross-chain communication process operations, the BSCTQCAT algorithm proposed in this paper outperforms the other two algorithms in terms of system efficiency through multiple runs of the test for fixed-size files. Therefore, the scheme in this paper can ensure the timeliness of the operation and the throughput of the system, and side by side, it highlights the stability.

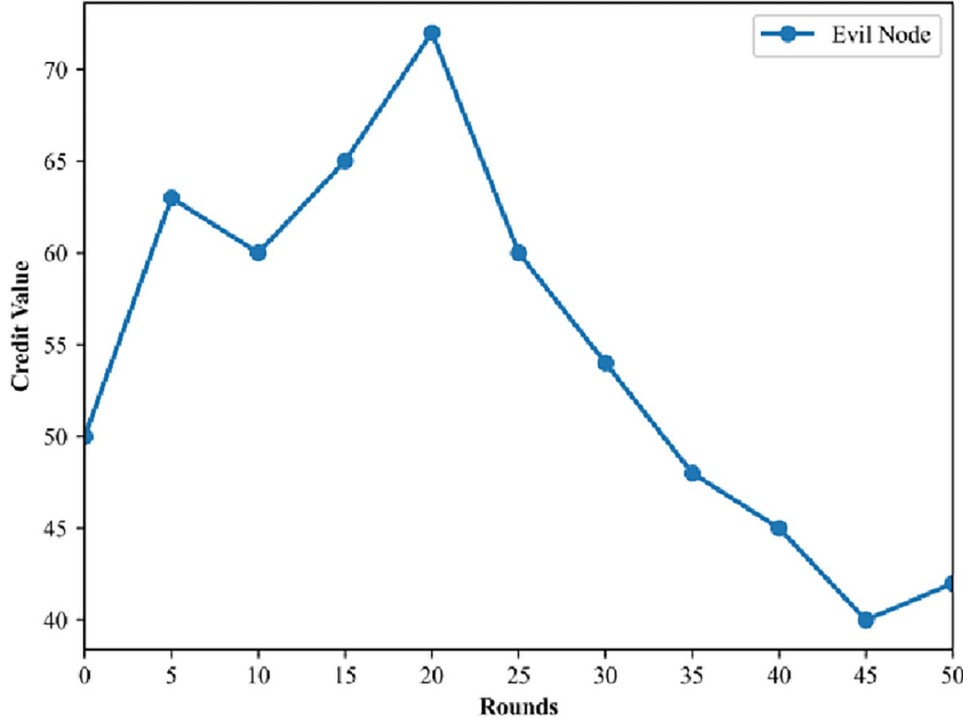

**Fig 5. Results of the 50 rounds of voting on the evil KG node.**

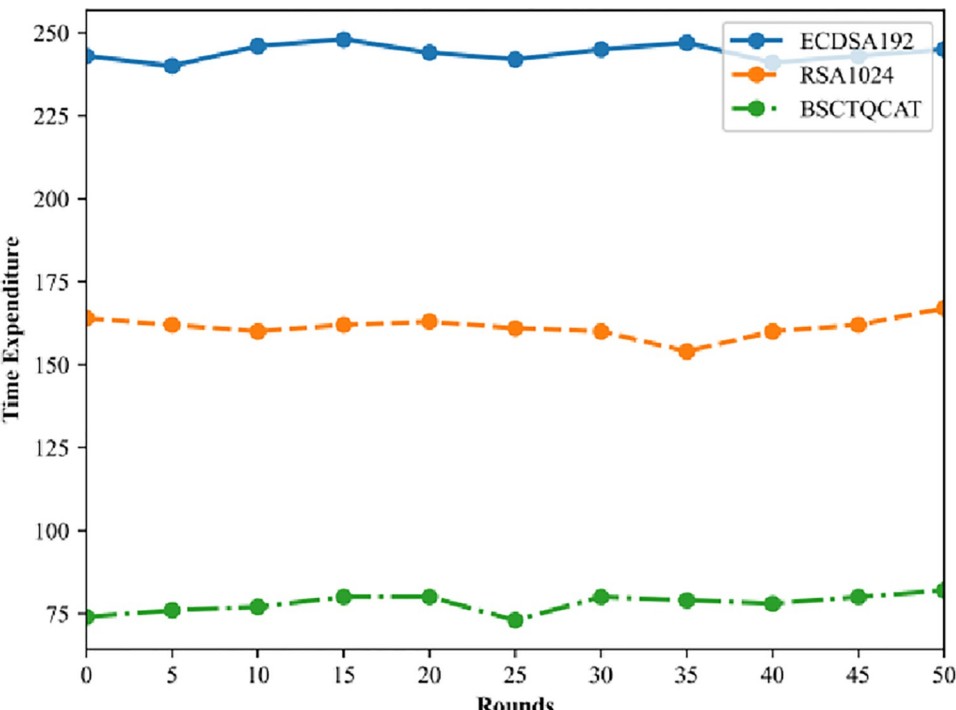

**Fig 6. Number of cross-chain communication process operations.**

## 7 Application scenario analysis

### 7.1 Application scenarios

A transaction is defined as an exchange of value between parties or multiple parties using currency as a medium. Transactions are realized by transferring digital assets representing monetary value from the sender's ledger to the receiver's ledger. As with traditional banking transactions, each transaction in the blockchain must clearly contain information such as the address where the transaction was generated, where the transaction was sent out, and the amount of the transaction. Through the transaction, a digital currency is created and the digital currency is transferred and tracked, which is one of the core features of the blockchain.

Each transaction in the blockchain requires a clear record of the sender, the receiver, the transaction amount and the associated transaction-generating address and transaction-output address information. This information makes up the transaction record, through which the transaction is monitored and traced, thus guaranteeing the security of the transaction. During the transaction process, the amount of digital currency will change with the transaction, and sometimes, it is necessary to carry out a change operation. The quantity and amount of the transaction, as well as the way to make change, need to be described and recorded in detail in the transaction record.

### 7.2 Analysis of case studies

**7.2.1 Case background.** The research in this paper uses Bitcoin as an application scenario, which is divided into two parts based on the transaction structure modeled on Bitcoin. The transaction structure mainly consists of an input part and an output part.

During the process of a transaction, the party initiating the transaction can be one or more accounts, while the party receiving the transaction can also be one or more accounts. In the input section, the information to be provided by the initiating party of the transaction includes the output address of the previous transaction, the amount of the transaction, a digital signature. The output portion represents the amount of the transaction that the receiver will receive and the address to which it will be deposited.

**7.2.2 Implementation strategies.** In a blockchain system, a wallet can be considered as an account which is composed of a signature mechanism, a public key, public key address mechanism. The digital signature mechanism is used to uniquely authenticate the transaction initiated by the account to ensure the authenticity and integrity of the transaction. The public key, on the other hand, is used to verify the validity and authenticity of that transaction during the transaction process. Meanwhile, the public key address is the address that must be used by other account nodes to transfer money to this account node.

In a wallet, the wallet holder creates his/her own public and private keys and associates them with his/her account using a digital signature. Other nodes in the network can then use this public key to validate transactions initiated by the account and confirm their authenticity. The public key address is a unique address derived from the data processing of the public key, which is used by other nodes to transfer money to the account, as shown in Fig 7. The wallet structure of the blockchain.

In addition to verifying the authenticity and uniqueness of the transaction, the signature mechanism and the public key address mechanism also guarantee the security of the transaction. The transaction is encrypted while the initiator of the transaction signs the transaction using a digital signature, thus guaranteeing the security of the transaction process.

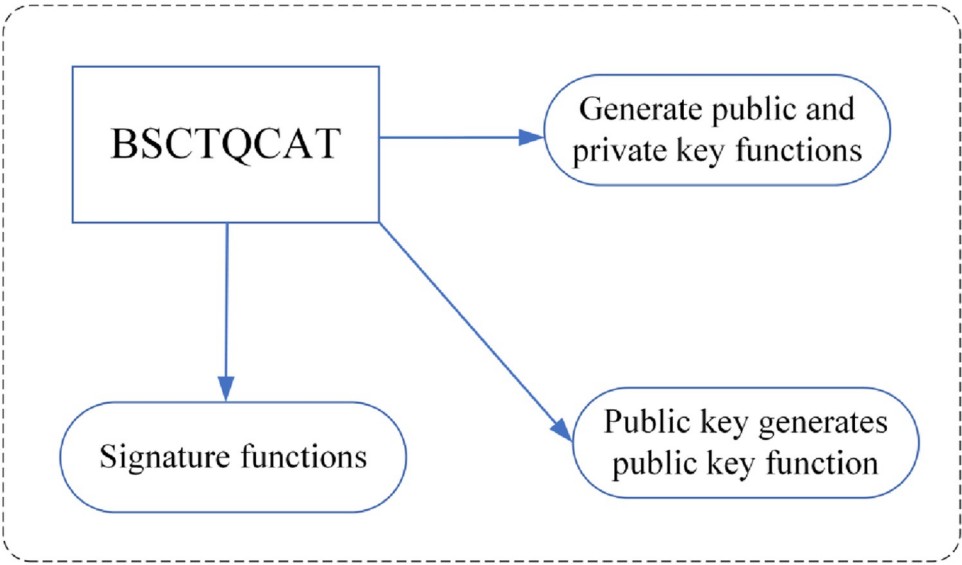

**Fig 7. The wallet structure of the blockchain.**

# 8 Conclusion

## 8.1 Major findings and conclusions

In this paper, a blockchain smart contract technology based on resistance to quantum computing attacks is proposed to address the cross-chain communication, data sharing and security challenges faced by current blockchain technology. The technique first introduces a digital signature scheme of lattice cyphers to improve the ability to resist quantum search algorithm attacks. Second, it uses a smart contract authentication scheme to organize nodes on multiple heterogeneous chains to construct an identity agent layer P2P network. Data sharing and establishing a trusted identity management and message authentication mechanism between different chains are realized through transactions, effectively facilitating cross-chain communication.

## 8.2 Limitations and perspectives

With the development of quantum computing technology, blockchain technology faces more and more severe security challenges. Therefore, the proposal in this paper is in line with the current technology trend. In future research, the standardization of cross-chain communication and data sharing also has to be deeply explored and promoted for promoting interoperability among different blockchains, achieving a larger range of data exchange to enhance the network effect, and further realizing the application of blockchain technology.

## Supporting information

**S1 File. Implementation and validation data for quantum-resistant signature algorithms.** (ZIP)

## Author Contributions

**Project administration:** Xinhao Zheng.

**Writing – original draft:** Xinhao Zheng.

**Writing – review & editing:** Xinhao Zheng.

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
