## [Decision Letter · Decision Letter 0]

9 Feb 2024

PONE-D-23-41869Research on Blockchain Smart Contract Technology Based on Resistance to Quantum Computing AttacksPLOS ONE

Dear Dr. Zheng,

Thank you for submitting your manuscript to PLOS ONE. After careful consideration, we feel that it has merit but does not fully meet PLOS ONE’s publication criteria as it currently stands. Therefore, we invite you to submit a revised version of the manuscript that addresses the points raised during the review process.

We look forward to receiving your revised manuscript.

Kind regards,

Chin-Ling Chen

Academic Editor

PLOS ONE

Journal Requirements:

Reviewers' comments:

Reviewer's Responses to Questions

**Comments to the Author**

1. Is the manuscript technically sound, and do the data support the conclusions?

Reviewer #1: Yes

Reviewer #2: Yes

2. Has the statistical analysis been performed appropriately and rigorously? 

Reviewer #1: N/A

Reviewer #2: Yes

3. Have the authors made all data underlying the findings in their manuscript fully available?

Reviewer #1: Yes

Reviewer #2: Yes

4. Is the manuscript presented in an intelligible fashion and written in standard English?

Reviewer #1: Yes

Reviewer #2: Yes

5. Review Comments to the Author

Reviewer #1: This paper is based on Research on Blockchain Smart Contract Technology Based on Resistance to Quantum Computing Attacks. Thus, this paper is directly related to the theme of this journal.

Overall, the paper is organized properly; the concept and future research directions are extensively explained. So, the paper is accepted after following minor changes:

1. Problem of paper and motivation is not clear in introduction

2. Comparison of current work is not given with previous research

3. Pseudo code of algorithm is given but comments of statements are not added for ease of readers

4. Paper contains few grammar mistakes which will be cooperated in final version.

5. Only few references are added in paper, but more than 70 references so to attract readers add few latest references related to this paper, which is mentioned below

Laghari, A. A., H. Shah, R. A. Laghari, K. Kumar, A. A. Waqan, and A. K. Jumani. "A Review on Quantum Computing Trends & Future Perspectives." EAI Endorsed Transactions on Cloud Systems 7, no. 22 (2022): e1-e1.

Khan, Abdullah Ayub, Sami Bourouis, M. M. Kamruzzaman, Myriam Hadjouni, Zaffar Ahmed Shaikh, Asif Ali Laghari, Hela Elmannai, and Sami Dhahbi. "Data Security in Healthcare Industrial Internet of Things with Blockchain." IEEE Sensors Journal (2023).

Laghari, Asif Ali, Abdullah Ayub Khan, Reem Alkanhel, Hela Elmannai, and Sami Bourouis. "Lightweight-BIoV: Blockchain Distributed Ledger Technology (BDLT) for Internet of Vehicles (IoVs)." Electronics 12, no. 3 (2023): 677.

Shaikh, Zaffar Ahmed, Abdullah Ayub Khan, Lin Teng, Asif Ali Wagan, and Asif Ali Laghari. "BIoMT modular infrastructure: The recent challenges, issues, and limitations in blockchain hyperledger-enabled e-healthcare application." Wireless Communications and Mobile Computing 2022 (2022): 1-14.

Khan, Abdullah Ayub, Zaffar Ahmed Shaikh, Asif Ali Laghari, Sami Bourouis, Asif Ali Wagan, and Ghulam Ali Alias Atif Ali. "Blockchain-aware distributed dynamic monitoring: a smart contract for fog-based drone management in land surface changes." Atmosphere 12, no. 11 (2021): 1525.

Shaikh, Zaffar Ahmed, Abdullah Ayub Khan, Laura Baitenova, Gulmira Zambinova, Natalia Yegina, Natalia Ivolgina, Asif Ali Laghari, and Sergey Evgenievich Barykin. "Blockchain hyperledger with non-linear machine learning: A novel and secure educational accreditation registration and distributed ledger preservation architecture." Applied Sciences 12, no. 5 (2022): 2534.

Khan, Abdullah Ayub, Zaffar Ahmed Shaikh, Larisa Belinskaja, Laura Baitenova, Yulia Vlasova, Zhanneta Gerzelieva, Asif Ali Laghari, Abdul Ahad Abro, and Sergey Barykin. "A blockchain and metaheuristic-enabled distributed architecture for smart agricultural analysis and ledger preservation solution: A collaborative approach." Applied Sciences 12, no. 3 (2022): 1487.

Reviewer #2: The author has proposed a new algorithm to counter quantum attacks on blockchain smart contracts. He has shown mathematical model that consider a vision of quantum computers.

It is desired to be considered to incorporate the following suggestions:

1. In 6.1 it needs to justify the reason to use give setup.

2. In 6.2, transactions are performed serially. Please mention that it will affect performance.

3. Fig. 7 should be shown with a direction sign,.

6. PLOS authors have the option to publish the peer review history of their article (what does this mean?). If published, this will include your full peer review and any attached files.

Reviewer #1: No

Reviewer #2: **Yes: **Sandeep Joshi

---

## [Author Response · Author response to Decision Letter 0]

28 Mar 2024

Dear editors and reviewers,

It is a great honor for me to have the opportunity to revise my manuscript entitled “Research on Blockchain Smart Contract Technology Based on Resistance to Quantum Computing Attacks”. Your insightful comments and valuable suggestions are highly appreciated and have been carefully addressed in the revision. Following are my detailed response to the specific comments. To facilitate the presentation, I have numbered the comments and responded one by one.

1.Response to Reviewer 1

Comment 1.1: Problem of paper and motivation is not clear in introduction.

Response 1.1: Thank you for your insightful comments. In the revised version, I have clarified the problem and the motivation for the research..

“However, with the development of quantum computing technology, existing digital signature algorithms, such as elliptic curve cryptography, lack resistance to quantum computing attacks[14-15]. Therefore, the research motivation of this paper is to address the cross-chain communication, data sharing, and security challenges faced by current blockchain technologies.”

Comment 1.2: Comparison of current work is not given with previous research.

Response 1.2: Thank you for your insightful comments. I compare the current work with previous studies.

“In the above research, existing digital signature algorithms, including elliptic curve encryption, face the risk of being cracked by quantum computing attacks. In the event that a quantum computing attack is able to successfully crack these algorithms, traditional digital signature mechanisms will no longer be secure. This issue will also have an impact on smart contracts, which typically use digital signatures to verify and authorize transactions. If quantum computing attacks crack traditional digital signature algorithms, the security of smart contracts will be jeopardized. Attackers potentially may be able to forge digital signatures or tamper with the execution results of smart contracts. Therefore, this paper proposed a blockchain smart contract technology based on anti-quantum computing attacks, which can effectively deal with quantum computing attacks and protect the security of smart contracts by adopting quantum-secure encryption algorithms and digital signature algorithms. In this way, the risk of quantum computing attacks is reduced, the security of digital signatures is ensured, which is not easy to be cracked or forged by attackers.”

Comment 1.3: Pseudo code of algorithm is given but comments of statements are not added for ease of readers.

Response 1.3: Thanks for your insightful comments. In the modified version, no statement comments were added. 

Algorithm 1: Trusted Identity Search

Input:BF={0}n, identifier[m]→ Array of query identity collections. id → Identity to be queried

i=0

// Loop over the m identity sets identifier to be queried

While(i<m){

j=0;

// For each identity identifier[i], compute the hash value by hj(identifier[i]) and set the corresponding BF array element to 1 to indicate that the identity may exist.

While(i<=k){

BF[ℎj(identifier[i])] = 1;

j=j+1;}

i=i+1;}

//Find id 

For i in range(0, k+1)：

// Iterate through the BF array, if there exists any BF array element of 0, it means the corresponding identity does not exist, return False

IF (hi(identifier))!=1：

Return False

//If all the BF array elements are 1, it means all the identities to be queried exist, return True.

Return True

Comment 1.4: Paper contains few grammar mistakes which will be cooperated in final version.

Response 1.4: I have fixed the grammatical errors in the revised version.

“In this paper, through the proposed technology based on anti-quantum computing attack, this paper successfully solves the risk of quantum computing attack faced by smart contracts and blockchain, providing a safer blockchain smart contract service.”is modified to

“In this paper, through the proposed technology based on anti-quantum computing attacks, this paper successfully solves the risk of quantum computing attacks faced by smart contracts and blockchain, providing a safer blockchain smart contract service. This is of great significance in promoting the development and application of blockchain technology.”

“These techniques ensure the security compliance of the business and automate and intelligentize the power business process, thus improving the efficiency of the business process.”is modified to

“These techniques ensure the security compliance of the business and automate and intelligentize the power of the business process, thus improving its efficiency.”

“The immutability of the blockchain ensures that once a smart contract is predefined, no changes are allowed.”is modified to“The immutability of the blockchain ensures that no changes are allowed once a smart contract is predefined.”

Comment 1.5: Only few references are added in paper, but more than 70 references so to attract readers add few latest references related to this paper, which is mentioned below.

Response 1.5: Thanks for your insightful comments. I have added the latest references related to this paper in the revised version.

[27] Khan A A, Shaikh Z A, Belinskaja L, et al. A blockchain and metaheuristic-enabled distributed architecture for smart agricultural analysis and ledger preservation solution: A collaborative approach[J]. Applied Sciences, 2022, 12(3): 1487.

[33] Shaikh Z A, Khan A A, Baitenova L, et al. Blockchain hyperledger with non-linear machine learning: A novel and secure educational accreditation registration and distributed ledger preservation architecture[J]. Applied Sciences, 2022, 12(5): 2534.

[40] Khan A A, Shaikh Z A, Laghari A A, et al. Blockchain-aware distributed dynamic monitoring: a smart contract for fog-based drone management in land surface changes[J]. Atmosphere, 2021, 12(11): 1525.

[47] Laghari A A, Khan A A, Alkanhel R, et al. Lightweight-biov: blockchain distributed ledger technology (bdlt) for internet of vehicles (iovs)[J]. Electronics, 2023, 12(3): 677.

[52] Khan A A, Bourouis S, Kamruzzaman M M, et al. Data Security in Healthcare Industrial Internet of Things with Blockchain[J]. IEEE Sensors Journal, 2023.

[61] Laghari A A, Shah H, Laghari R A, et al. A Review on Quantum Computing Trends & Future Perspectives[J]. EAI Endorsed Transactions on Cloud Systems, 2022, 7(22): e1-e1.

[64] Shaikh Z A, Khan A A, Teng L, et al. BIoMT modular infrastructure: The recent challenges, issues, and limitations in blockchain hyperledger-enabled e-healthcare application[J]. Wireless Communications and Mobile Computing, 2022, 2022: 1-14.

2.Response to Reviewer 2

Comment 2.1: In 6.1 it needs to justify the reason to use give setup.

Response 2.1: Thank you for your insightful comments. In the revised version, I give the reason for using the given settings.

“The effectiveness of the scheme proposed in this paper is tested through simulation experiments to verify whether the scheme proposed in this paper is capable of meeting the resistance to quantum computation in cross-chain communication scenarios.”

Comment 2.2: In 6.2, transactions are performed serially. Please mention that it will affect performance.

Response 2.2: Thank you for your insightful comments. I have elaborated on this in the revision.

Table 2 Time for each phase of a digitally signed blockchain transaction

Stage Average time Standard deviation of time

Signature 0.035 0.003

Verification 0.184 0.004

Transaction 0.235 0.006

According to the results in Table 2, the duration of each phase are all at the microsecond level, which means that the performance of the whole blockchain system system is less affected by the fact that the transactions are done in a "serial" way. However, the serial approach clearly shows the specific flow of each transaction, including the signature and verification process, which is easy to understand and debug. Each transaction is executed in a precise sequence without the complexity and competitive conditions associated with concurrent operations.

Comment 2.3: . Fig. 7 should be shown with a direction sign.

Response 2.3: Thank you for your insightful comments. I have modified Figure 7.

Fig.7 The wallet structure of the blockchain

---

## [Editor Report · Decision Letter 1]

2 Apr 2024

Research on Blockchain Smart Contract Technology Based on Resistance to Quantum Computing Attacks

PONE-D-23-41869R1

Dear Dr. Zheng,

We’re pleased to inform you that your manuscript has been judged scientifically suitable for publication and will be formally accepted for publication once it meets all outstanding technical requirements.

Kind regards,

Chin-Ling Chen

Academic Editor

PLOS ONE
---

## [Editor Report · Acceptance letter]

10 May 2024

PONE-D-23-41869R1 

PLOS ONE

Dear Dr. Zheng, 

I'm pleased to inform you that your manuscript has been deemed suitable for publication in PLOS ONE. Congratulations! Your manuscript is now being handed over to our production team.

Kind regards, 

on behalf of

Prof. Chin-Ling Chen 

Academic Editor

PLOS ONE